# When Therapy-Induced Cancer Cell Apoptosis Fuels Tumor Relapse

**Razmik Mirzayans**

Department of Oncology, University of Alberta, Cross Cancer Institute, Edmonton, AB T6G 1Z2, Canada; razmik.mirzayans@ahs.ca

**Simple Summary:** For nearly half a century, a mainstay and goal of medical oncology has been to identify novel anticancer drugs and therapeutic strategies to promote the effective elimination of cancer cells via apoptosis. In the past decade, however, single cell biology has revealed that apoptosis is not obligatorily a permanent cell fate. The purpose of this commentary is to briefly discuss laboratory and clinical studies that have revealed the dark side of apoptosis in treating patients with solid tumors.

**Abstract:** Most therapeutic strategies for solid tumor malignancies are designed based on the hypothesis that cancer cells evade apoptosis to exhibit therapy resistance. This is somewhat surprising given that clinical studies published since the 1990s have demonstrated that increased apoptosis in solid tumors is associated with cancer aggressiveness and poor clinical outcome. This is consistent with more recent reports demonstrating non-canonical (pro-survival) roles for apoptotic caspases, including caspase 3, as well as the ability of cancer cells to recover from late stages of apoptosis via a process called anastasis. These activities are essential for the normal development and maintenance of a healthy organism, but they also enable malignant cells (including cancer stem cells) to resist anticancer treatment and potentially contribute to clinical dormancy (minimal residual disease). Like apoptosis, therapy-induced cancer cell dormancy (durable proliferation arrest reflecting various manifestations of genome chaos) is also not obligatorily a permanent cell fate. However, as briefly discussed herein, compelling pre-clinical studies suggest that (reversible) dormancy might be the "lesser evil" compared to treacherous apoptosis.

**Keywords:** solid tumor therapy; therapy resistance; caspase 3; apoptosis; anastasis; senescence; intratumor heterogeneity

## 1. Introduction

The theme of the AACR meeting held over two decades ago (February 2002) in Hawaii was "Apoptosis and Cancer: Basic Mechanisms and Therapeutic Opportunities in the Post-Genomic Era". Gozani et al. [1] published the meeting report, entitled "Death in Paradise", with the following opening: "Picture a starry sky, luminous waves, throbbing drums, the smouldering smell of roast pig, and a huddle of scientists discussing death. Not exactly paradise, at least for the pig, but that is what we experienced at this year's AACR (conference)...this year's speakers enthralled attendees with their recent discoveries in the field of apoptosis and cancer research". The authors concluded the report by stating that "...the field cannot pause to congratulate itself on its accomplishments, for we are still far from developing a 'magic bullet' capable of harnessing our knowledge of apoptotic pathways to eradicate cancers. We are hopeful that the knowledge of apoptosis and cancer that we possess today will be translated into new and more effective cancer therapies in the next decade".

Unfortunately, after decades of extensive research and clinical trials, novel apoptosis-triggering therapeutic strategies under the term "precision oncology" still remain to fulfill

their promises (reviewed in, e.g., [2–11]). In fact, a brief review of the history of cancer research has revealed that modern strategies for treating patients with certain types of solid tumors (e.g., esophageal cancer) may cause more harm than benefit (reviewed in [10]). This is in part because apoptosis-promoting therapy fuels "the oncogenic fire" ([12,13]; see also Figure 1).

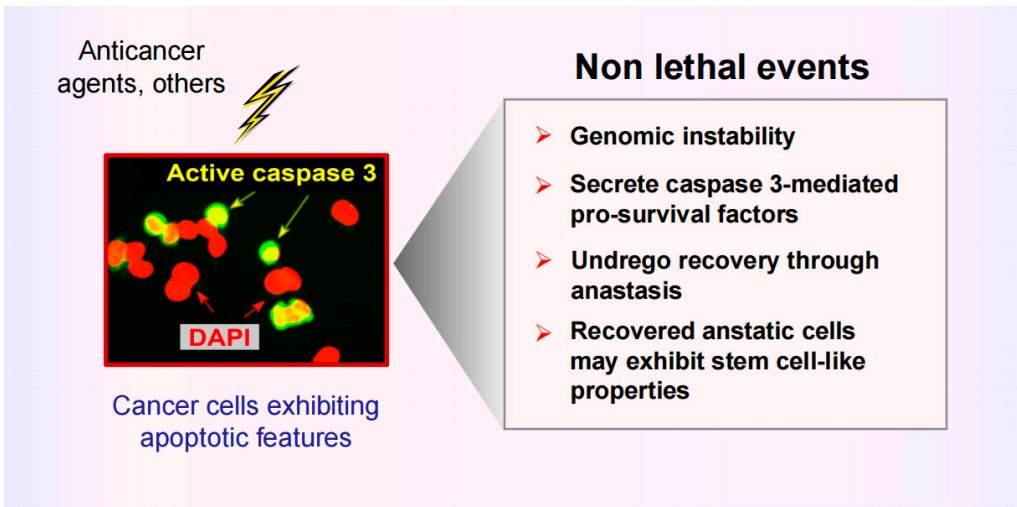

**Figure 1.** Cartoon illustrating the dark side of apoptosis in solid tumor therapy. Cancer cells triggered to undergo apoptosis are capable of promoting tumor repopulation via different routes, including caspase 3-mediated secretion of pro-survival factors and the ability to return from various stages of apoptosis (even after formation of apoptotic bodies), which can result in the emergence of aggressive variants. Adapted from Mirzayans and Murray [13].

This commentary highlights laboratory and clinical studies that have revealed the dark side of apoptosis in the context of cancer therapy and is meant to be complementary to recent reviews by pioneers in the field of regulated cell death [14–16]. Similar to our previous reports (e.g., [10,11]), the conclusions drawn from the studies outlined herein pertain to solid tumors/tumor-derived cell lines which may or may not be applicable to hematologic malignancies.

Apoptosis is the most extensively studied form of regulated (programmed) cell death involving an energy-dependent cascade of molecular events [17–19]. There are two main pathways of apoptosis: the extrinsic (or death receptor) pathway and the intrinsic (or mitochondrial) pathway. Lethal stimuli such as anticancer agents activate the intrinsic pathway. The process is initiated by mitochondrial outer membrane permeabilization (MOMP), resulting in the release of cytochrome c and other intermembrane space proteins. Cytochrome c interacts with apoptosis-activating factor-1 (APAF-1), which recruits pro-caspase 9 to form the apoptosome. This interaction results in the activation of caspase 9, which in turn activates the executioner caspases (caspase 3 and caspase 7). The execution phase leading to cell demise has been recently referred to as "apoptosis" by the Nomenclature Committee on Cell Death [18] and others (e.g., [19]). This phase includes the externalization of phosphatidylserine (PS) on the outer plasma membrane leaflet, internucleosomal DNA cleavage, nuclear condensation, cell shrinkage, and the eventual formation of apoptotic bodies [17]. For further details, see [17–19].

Up until a decade ago, we had assumed that cancer cell apoptosis triggered by ionizing radiation, chemotherapeutic drugs, and other stimuli will inevitably lead to their demise. Accordingly, like many (thousands) authors, we had considered cancer cell apoptosis to represent a favorable clinical outcome for the treatment of patients with solid tumors [20]. This was consistent with the hypothesis that evasion of apoptosis might represent a fundamental trait ("hallmark") of cancer, underlying therapy resistance and relapse, as proposed

by Hanahan and Weinberg in their seminal paper published in 2000 [21]. As it turned out, we were wrong!

In a comprehensive book chapter entitled "The Dark Side of Apoptosis" published in 2013 [22], Malathy Shekhar discussed accumulating evidence for the paradoxical role of apoptosis in tumor progression. In breast cancer, for example, BCL2 overexpression was already (over a decade ago) known to be associated with "normal ploidy, estrogen receptor positivity, and absence of metastasis; all characteristics associated with better clinical outcome and a more favorable prognosis that is contradictory to its (BCL2's) predicted role in apoptosis resistance" [22]. As pointed out by this author [22] and others (e.g., [23–26]), numerous clinical studies published since the mid-1990s have shown that patients with high rates of apoptosis had significantly worse prognosis compared to patients with low apoptotic rates; the study reported by Yang et al. in 2018 [26] involved a meta-analysis of 3091 breast cancer cases. These clinical observations do not support the aforementioned popular model proposed by Hanahn and Weinberg [21] (see also Section 2 below).

In the past decade, the number of studies underscoring the dark side of apoptosis in the context of treating patients with solid tumors has grown at a rapid pace, some of which are outlined below (see also Figure 1).

Caspases are proteases with a well-defined role in apoptosis, with caspase 3 functioning as the principal executioner caspase in both intrinsic and extrinsic apoptotic pathways. Increasing evidence, however, indicates multiple functions of caspases outside apoptosis. Caspase 3, for example, which has often been considered as a reliable marker for cancer cell death and thus efficacy of cancer therapy, paradoxically plays key roles in promoting the survival and proliferation of malignant cells [27–33]. The complex pro-survival, tumor-repopulating process associated with dying (active caspase 3-expressing) cells is termed "Phoenix Rising" [27–31], from the mythical bird re-born from its own ashes, or "Failed Apoptosis" [12,13,32], or "Treacherous Apoptosis" [33].

The dark side of apoptosis is not limited to the oncogenic function of caspase 3. In the early 2000s, Geske et al. [34,35] reported that mammalian cells can recover from at least early stages of apoptosis after removal of the apoptotic stimulus, suggesting that apoptosis is not obligatorily a permanent cell fate. This fundamental discovery went largely unnoticed except for occasional mentions in review articles (e.g., [17]). Since 2009, however, Tang et al. [36] and numerous other groups have independently reported such a recovery phenomenon in different biological systems, including cancer stem cells [37–39] and solid tumor-derived cell lines treated with anticancer agents (review in, e.g., [14–16,40–43]). Cancer cells can return from not only the early stages of apoptosis (caspase activation) but also after mitochondrial fragmentation, nuclear condensation, cell shrinkage, and apoptotic body formation [38,39,43]. The return journey from engaging apoptosis (and other modes of cell death such as ferroptosis [44]) is now being referred to as anastasis (Greek for "rising to life"). (The process of cell survival after engaging necroptosis is termed resuscitation; reviewed in [16]).

Anastasis in cancer cells results in the emergence of progeny with an increased number of micronuclei and chromosomal abnormalities that can lead to increased aneuploidy [43,45], a driving force of aggressive cancer [7]. The molecular basis for anastasis-driven tumor angiogenesis and metastasis is emerging. Three recent reports from the laboratory of Gongping Sun, for example, have demonstrated roles for cIAP2/NFκB [46], CDH12 [47], and p38 MAPK signaling [48] in these processes.

Different cancer cell fate outcomes after engaging apoptosis and other modes of regulated cell death contribute to intertumor heterogeneity (differences in terms of therapy response between patients with the same type/stage of cancer), as well as intratumor heterogeneity (distinct tumor cell populations exhibiting therapy resistance through different mechanisms). The ever-increasing complexity that exists within a solid tumor poses a major challenge in implementing precision oncology. This complexity has been recently reviewed by us [11,49] and other groups [50–53] (see also Appendix A).

In addition to regulated cell death, therapeutic agents trigger cancer cell dormancy (durable, but often reversible proliferation arrest) via different routes, including stress-induced premature senescence (also called therapy-induced senescence). The mechanisms underlying the reversal of dormancy in solid tumors/tumor-derived cell lines, potentially resulting in therapy-resistance and disease recurrence, have been well documented and extensively reviewed (e.g., [54–58]). A fairly recent article entitled "Tumor Cell Senescence Response Produces Aggressive Variants" by Yang et al. [59] is of particular relevance to the current discussion as it illustrated another potentially dark side of apoptosis in cancer therapy. The authors determined the impact of ectopic expression of caspase 3 or treatment with apoptosis-inducing drugs (e.g., camptothecin; the BCL2 inhibitor ABT-737) on the fate of senescent lung carcinoma (e.g., A549) and breast carcinoma (e.g., MCF7) cells. Triggering apoptosis in senescent cancer cells was shown to accelerate reversal of the proliferation arrested state rather than leading to their demise.

As we have extensively discussed previously [49,57,58], numerous studies reported in the past three decades, including our own work published since the 1990s [60–63], have established the presence of a threshold mechanism for stress-induced apoptosis in most human cell types. Importantly, studies with solid tumors and solid tumor-derived cell lines have demonstrated that a major response triggered by low/moderate doses of DNA-damaging agents (ionizing radiation, chemotherapeutic drugs administered under clinically relevant conditions) is a sustained proliferation arrest (dormancy), rather than apoptosis. The cartoon in Figure 2 is reproduced from our 2016 review [57]; it illustrates the apoptotic threshold in p53 wild-type HCT116 human colon cancer cells after treatment with the chemotherapeutic drugs cisplatin and oxaliplatin. The presence of an apoptotic threshold is consistent with the antiapoptotic properties of p53 [64,65] and its transcriptional targets p21$^{WAF1}$ (p21) [64–66], WIP1 [64–67], and others [64,65].

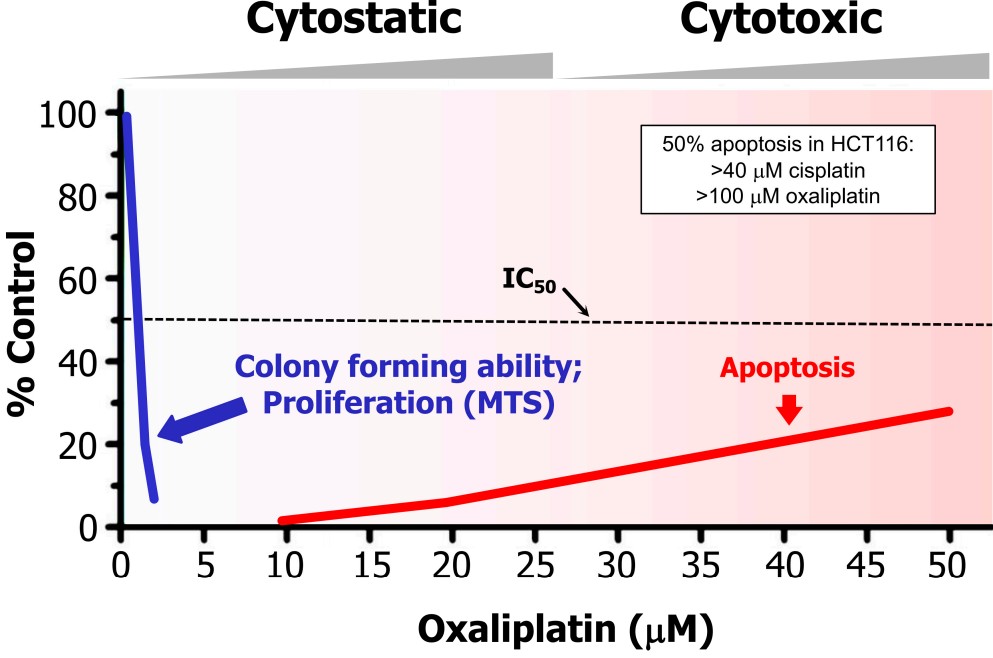

**Figure 2.** Dose-dependent responses induced by oxaliplatin in p53 wild-type HCT116 colon carcinoma cells, measured by colony formation, cell proliferation (e.g., MTS), and apoptosis assays (for details, please see [57]). Such large discrepancy between drug concentrations required to induce a cytostatic/dormancy response (durable proliferation arrest) versus apoptosis has also been reported for this and other solid tumor-derived cell lines after treatment with cisplatin. In HCT116 cells, for example, <5 μM and >40 μM cisplatin concentrations induced 50% effect (IC$_{50}$) in colony formation and apoptosis assay, respectively. Reproduced from [57].

In view of this nonlinear dose relationship in cell fate outcome, it is reasonable to assume that cancer cell dormancy might be the "lesser evil" than treacherous apoptosis, simply because relatively low/moderate doses of anticancer agents required to trigger dormancy (e.g., via senescence) will undoubtedly cause less unwanted side effects, such as compromising patient's immune system, as compared to high doses required to induce apoptosis.

Given that triggering cancer cell dormancy following conventional cancer therapies is almost inevitable, some authors have suggested a 'one–two punch' strategy in which the sequential treatment of cancer with pro-senescence therapy is followed by "senolytic" therapy [68,69]. Time will tell whether such an approach can be used to significantly improve the length and quality of life of cancer patients. It would be unrealistic to predict cancer cure by implementing such an approach when considering the degree of complexity that exists within a tumor that can underlie resistance and relapse (intratumor heterogeneity) [11,49] (see also Appendix A). For example, a subset of cancer cells within a tumor/tumor-derived cell line responds to therapeutic agents by exhibiting various manifestations of genome chaos [7], resulting in the creation of polyploid/multinucleated giant cancer cells (PGCCs), which have emerged as the root causes of therapy resistance and disease relapse (reviewed in [11,55,56,70]). PGCCs as well as other therapy resistant sub-populations within a tumor can emerge via different routes, including non-mutational mechanisms such as cell fusion (reviewed in [11,71]).

## 2. Conclusions

Extensive preclinical and clinical studies reported following the aforementioned AACR meeting on "Apoptosis and Cancer" [1] have revealed a grim reality: apoptosis is not the end-point of anticancer treatment, but rather represents a key turning point of unwanted side effects triggered by cytotoxic (proapoptotic) therapies, with the result that the initial benefits of tumor shrinkage are overwhelmed by a successive exaggerated tumor repopulation. (This narration was adapted from Corsi et al. [30]). Fortunately, ionizing radiation and low/moderate doses of conventional chemotherapeutic agents predominantly trigger cancer cell dormancy, which appears to be a more favorable clinical outcome than "treacherous" apoptosis.

As recently (2024) pointed out by Nano and Montell in a comprehensive article entitled "Apoptotic signaling: Beyond cell death" [15], the following fundamental question remains to be addressed by the apoptosis/anastasis community: "what is the point of no return in apoptotic commitment?" Addressing this question is important for assessing the validity of radiosensitivity and chemosensitivity data obtained by widely used cell death "markers" (e.g., caspase activation, MOMP, PS externalization, nuclear fragmentation, etc.).

Alternative explanations for cancer cell survival after engaging apoptosis cannot be ruled out. In fact, studies demonstrating that cancer cells can recover from late stages of apoptosis, even after the formation of apoptotic bodies [38,39,43], argue against the presence of a point of no return in this mode of cell death. Perhaps as long as antiapoptotic factors such as p21 are present in a cell, the cell may have the opportunity to survive after being triggered to undergo apoptosis. (p21 can be induced via p53-dependent and -independent mechanisms [72]). In other words, perhaps apoptotic cancer cells face a "molecular brick wall" that prevents their demise, rather than needing to evade an elusive point of no return to survive.

Another fundamental question arises after considering the discoveries discussed in this commentary. Namely, is "Evading Apoptosis" a hallmark of cancer, contributing to therapy resistance, as hypothesized by Hanahan and Weinberg in 2000 [21], or cancer cells simply employ homeostatic processes (e.g., anastasis; caspase-mediated proliferation) to survive after engaging apoptosis and other modes of regulated cell death? To this end, it is important to note that the properties of p53 and its downstream effectors (p21, WIP1, and numerous others [64–66]) are not consistent with the "Evading Apoptosis" model. As alluded to earlier, these key mediators of the DNA damage response serve to

suppress apoptosis rather than promote it. Thus, inhibiting apoptosis might be considered to represent the "bright" side of a tumor suppressor (p53) rather than its "dark" side, as proposed by Jänicke et al. [64] in 2008.

Food for thought: In 2016 Vinay Prasad published a perspective article in *Nature* entitled "The precision-oncology illusion" in which he argued that "Precision oncology has not been shown to work, and perhaps it never will. . .we may expect short-lived responses in a tiny fraction of patients, with the inevitable toxicity of targeted therapies and inflated cost that this approach guarantees" [2]. Discoveries highlighted in the current article and previously (e.g., [10,11]) will hopefully enable the reader to elaborate or debate on the conclusion reached by Prasad [2] as well as others who have referred to personalized/precision oncology for the treatment of solid tumor malignancies as "failed medicine" or (empty) promises that remain to be fulfilled [3–9,73–76].

**Funding:** This research received no external funding.

**Institutional Review Board Statement:** Not applicable.

**Informed Consent Statement:** Not applicable.

**Conflicts of Interest:** The author declares no conflicts of interest.

**Appendix A**

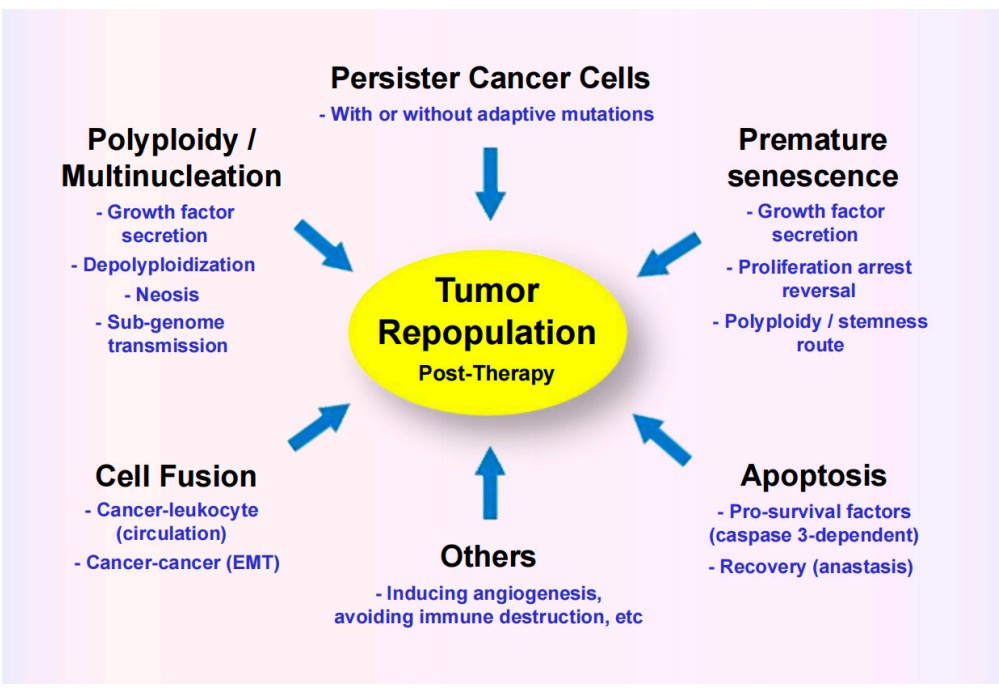

**Figure A1.** Responses contributing to intratumor heterogeneity, with different sub-populations of cancer cells within the same solid tumor exhibiting therapy resistance via different mechanisms. EMT, epithelial to mesenchymal transition. Reproduced from Mirzayans and Murray [49].

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
