# Peer review of "When Therapy-Induced Cancer Cell Apoptosis Fuels Tumor Relapse"

_onco, doi:10.3390/onco4010003_

Round 1
Reviewer 1 Report
Comments and Suggestions for Authors
The Commentary written by Dr. Mirzayans submitted for consideration for publication in Onco. The topic is interesting, and the commentary is well-written, similarly to many other articles published by this author. Although the idea that apoptosis is not the best way to eliminate solid tumor cells is not novel, the author discusses several important observations and accumulating evidence for the unexpected role of apoptosis in tumor progression. It is clear that the topic of this commentary is within the scope of Onco and will be interesting for the readers of the Journal. However, before acceptance the author should address several concerns.
1. Page 2. Indeed, in the “Millenium review” (2000) Hanahan and Weinberg suggested that evasion of apoptosis might represent one of the hallmarks of cancer. In fact, it was not wrong. At that time not too much was known about other modes of cell death and in the second edition (2011) "apoptosis" was substituted by “cell death”. So, this should be clarified. Moreover, the statement (ref. 21) in 2014(!) (after publication of “Next generation” in 2011) is not correct and should not be used as an argument.
2. Figure 1. Cytochrome c binds APAF-1, but not caspase-9, which is written.
3. Figure 1. Caspase-9 indeed directly activates caspase-3/-7, but not DNase activity.
4. During “non-lethal” events apoptosis continues to develop, therefore, the regions within a tumor are still caspase 3-positive. However, the intensity of anastasis is much higher, which leads to accumulation of tumor cells and looks as “failed apoptosis”. Apoptosis occurs, but very slowly, which manifests itself in even greater tumor development.
5. The statement in Conclusion, that “the dark side of apoptosis and other modes of cell death (e.g., ferroptosis [41]) need to be taken into consideration when designing novel therapeutic strategies” should be expanded in order to avoid misunderstanding of the whole process.
6. There are several misprints that should be omitted.
Author Response
GENERTAL COMMENTS:
Many thanks to the three reviewers who provided valuable suggestions. Red font is used for revised/new text. Figures 1 and Appendix A are new, which I think better highlight the purpose of this commentary. The misprints are corrected and the description of the mechanism of apoptosis is corrected.
The suggestion of reviewer 3 that “Adding more opinions from the author to the paper might have significance for the readers” prompted me to expand the Conclusions section and raise some fundamental questions (together with my opinions) for the reader to elaborate or debate.
Two reviewers have indicated that the “English language is fine. No issues detected” whereas reviewer 2 has indicated that “Moderate editing of English language is required.” I assume that this is due to some misprints and errors (e.g., in apoptosis description), which are now corrected.
The ONCO editorial office has highlighted a number of sentences and asked me to try to rewrite them to reduce overlap with published work. I have used quotation marks when I have meant to copy a statement from published papers; some of my sentence structures in the current paper are similar to my own published material, but they are not identical, otherwise I would have put them in quotation marks.
REPLY TO REVIEWER 1 COMMENTS:
Point #1: The indicated sentence (Thus, “resistance to apoptosis should not be taken as a hallmark of cancer” [21]) is deleted so that it is not taken as an evidence to challenge hallmark #3. My assessment regarding this hypothesis, which was proposed in 2000, is more clearly described in the revised version. To this end, section 2 is expanded and draws some conclusions regarding this hallmark. Although it is not the main purpose of this commentary, my intention is to let the reader know about compelling solid data (published since the 1990’s) that challenge the hypothesis that “apoptosis evasion” underlies therapy resistance of solid tumor malignancies. (The fact that highly simplistic hypotheses such as this have derailed cancer research for decades will be extensively discussed in my next, and perhaps final review, if I find the drive to write it.)
Points #2 and #3: Thank you for highlighting these errors.
Points #4: This is an interesting scenario which needs to be tested, and I decided not to comment on in the current commentary. One of the reasons is that studies discussed in ref 33 (“Treacherous apoptosis—Cancer cells sacrifice themselves at the altar of heterogeneity”) demonstrating the presence of apoptosis-rich regions (pockets) within a tumor, which is associated with survival, was performed with hepatocellular carcinoma only, and did not address the relation to anastasis. Undoubtedly this work will be expanded to illustrate the generality of these intriguing discoveries. I thank the reviewer for this interpretation; I will think about it and may discuss it in the aforementioned review that I’m planning to work on next.
Points #5: This sentence is removed and replaced with more pertinent conclusions.
Points #6: Misprints are corrected, including a major one in the beginning of the Introduction (wrong authors for ref. 1).
Reviewer 2 Report
Comments and Suggestions for Authors
the manuscript submitted tackeles an interesting topic but could be improved in several aspects such as having a figure for the mechanism of apoptosis, summarizing figure for the pre-clinical studies mentioned in the submitted version.
More focus should be put on the heterogenous nature of the tumors from the inter-tumor and intre-tumor heterogenity presppective
Comments on the Quality of English LanguageMinor mistakes
Author Response
GENERTAL COMMENTS:
Many thanks to the three reviewers who provided valuable suggestions. Red font is used for revised/new text. Figures 1 and Appendix A are new, which I think better highlight the purpose of this commentary. The misprints are corrected and the description of the mechanism of apoptosis is corrected.
The suggestion of reviewer 3 that “Adding more opinions from the author to the paper might have significance for the readers” prompted me to expand the Conclusions section and raise some fundamental questions (together with my opinions) for the reader to elaborate or debate.
Two reviewers have indicated that the “English language is fine. No issues detected” whereas reviewer 2 has indicated that “Moderate editing of English language is required.” I assume that this is due to some misprints and errors (e.g., in apoptosis description), which are now corrected.
The ONCO editorial office has highlighted a number of sentences and asked me to try to rewrite them to reduce overlap with published work. I have used quotation marks when I have meant to copy a statement from published papers; some of my sentence structures in the current paper are similar to my own published material, but they are not identical, otherwise I would have put them in quotation marks.
REPLY TO REVIEWER 2 COMMENTS:
Figure 1 is replaced with the one which illustrates the purpose of this commentary right upfront. The text on apoptosis is corrected and expanded to provide sufficient information in the context of this commentary; for further details on apoptosis the reader is referred to recent (2023) reviews. I tried to include a figure for apoptosis mechanisms, but decided not to because this is not a review on apoptosis, but rather on its dark side (which stands out in revised figure 1).
Based on the suggestion of this reviewer, a new paragraph is added (end of page 3) concerning tumor heterogeneity with an emphasis on the contributions of responses that are the focus of this commentary, and for further details the reader is directed to recent reviews by us and others on tumor heterogeneity. The original Figure in Appendix A on intratumor heterogeneity is also replaced with a more informative one, which is a revised version from our 2020 review (ref. 49).
Reviewer 3 Report
Comments and Suggestions for Authors
This manuscript overviewed the dark side of apoptosis in treating patients with solid tumors. This Reviewer deem that the contents in the manuscript are important and interesting. The paper is well written and valuable for publication. Adding more opinions from the author to the paper might have significance for the readers.
Author Response
GENERTAL COMMENTS:
Many thanks to the three reviewers who provided valuable suggestions. Red font is used for revised/new text. Figures 1 and Appendix A are new, which I think better highlight the purpose of this commentary. The misprints are corrected and the description of the mechanism of apoptosis is corrected.
The suggestion of reviewer 3 that “Adding more opinions from the author to the paper might have significance for the readers” prompted me to expand the Conclusions section and raise some fundamental questions (together with my opinions) for the reader to elaborate or debate.
Two reviewers have indicated that the “English language is fine. No issues detected” whereas reviewer 2 has indicated that “Moderate editing of English language is required.” I assume that this is due to some misprints and errors (e.g., in apoptosis description), which are now corrected.
The ONCO editorial office has highlighted a number of sentences and asked me to try to rewrite them to reduce overlap with published work. I have used quotation marks when I have meant to copy a statement from published papers; some of my sentence structures in the current paper are similar to my own published material, but they are not identical, otherwise I would have put them in quotation marks.